# Niacin Status Indicators and Their Relationship with Metabolic Parameters in Dairy Cows during Early Lactation

**DOI:** 10.3390/ani12121524

**Published:** 2022-06-12

**Authors:** Kosta Petrović, Radojica Djoković, Marko Cincović, Talija Hristovska, Miroslav Lalović, Miloš Petrović, Mira Majkić, Maja Došenović Marinković, Ljiljana Anđušić, Gordana Devečerski, Dragica Stojanović, Filip Štrbac

**Affiliations:** 1Department of Veterinary Medicine, Faculty of Agriculture, University of Novi Sad, 21000 Novi Sad, Serbia; kostapetrovic84@gmail.com (K.P.); mcincovic@gmail.com (M.C.); miramajkic@gmail.com (M.M.); maja_2511@yahoo.com (M.D.M.); dragicas@polj.edu.rs (D.S.); strbac.filip@gmail.com (F.Š.); 2Faculty of Agronomy, University of Kragujevac, 32000 Čačak, Serbia; petrovic.milos87@yahoo.com; 3Veterinary Faculty, University of St. Kliment Ohridski, 7000 Bitola, North Macedonia; talijahris@gmail.com; 4Faculty of Agriculture East Sarajevo, University of East Sarajevo, 71123 East Sarajevo, The Republic of Srpska, Bosnia and Herzegovina; miroslav.lalovic@pof.ues.rs.ba; 5Faculty of Agriculture, University of Priština, 38219 Lešak, Serbia; lunaa.ns@gmail.com; 6Faculty of Medicine, University of Novi Sad, 21000 Novi Sad, Serbia; gordana.devecerski@mf.uns.ac.rs

**Keywords:** cows, niacin, active form, metabolic profile, early lactation

## Abstract

**Simple Summary:**

The active forms of niacin that represent niacin status are nicotinamide adenine dinucleotide (NAD), nicotinamide adenine dinucleotide phosphate (NADP) and the NAD:NADP ratio. Previous studies have shown metabolic changes in the function of niacin form and dose, but it has not been determined whether there are changes in the function of active form of niacin that indicate the vitamin status in the body. In this study, we examined differences in NAD, NADP and NAD:NADP concentration in blood and their relationship with metabolic parameters in cows receiving and not receiving additional niacin in food. We concluded that NAD and NADP are good indicators of the ability of an additional niacin source to create functional cofactors due to their concentration changes, while the NAD:NADP ratio is a good indicator of the biological effects of additional niacin due to correlation with many metabolites.

**Abstract:**

Previous experimental models on cows have examined the difference in the metabolic adaptation in cows after niacin administration, without identifying the most important mediators between niacin administration and its biological effects, namely active forms of niacin. All tissues in the body convert absorbed niacin into its main metabolically active form, the coenzyme nicotinamide adenine dinucleotide (NAD) and nicotinamide adenine dinucleotide phosphate (NADP). The aim of this study was to determine the influence of niacin administration in periparturient period on NAD, NADP and the NAD:NADP ratio and to determine relationship between these indicators of an active form of niacin with metabolic parameters in cow blood. The study included 90 healthy cows: 45 cows receiving niacin and 45 cows were negative control. The niacin group was treated with nicotinic acid for two weeks before, as well as two weeks after parturition. Nicotinic acid was applied per os with feed. In cows receiving niacin, there was a significantly higher concentration of NAD and NADP, but the NAD:NADP ratio did not differ compared with control. All three indicators were able to separate cows who received and who did not receive additional niacin. NAD and NADP are good indicators of the availability of niacin from additional sources. The NAD:NADP ratio is a good indicator of the biological effect of applied niacin on metabolites in cows due to its correlation with a number of metabolites: positive correlation with glucose, insulin, glucose to insulin ratio and the revised quantitative insulin sensitivity check index (RQUICKIBHB) of insulin resistance, triglycerides and cholesterol, and a negative correlation with nonesterified fatty acid (NEFA), beta hydroxybutyrate (BHB), gamma-glutamyltranspherase (GGT) and urea in cows receiving niacin. The same amount of added niacin in feed can produce different concentrations of NAD, NADP and NAD:NADP in the blood, and this was not related to their concentration before the addition of niacin. The change in the concentration of the active form of niacin (NAD, NADP and NAD:NADP) further correlates with the concentration of metabolic parameters, which indicates that the intensity of the biological effect of additional niacin can be accurately determined only if we know the concentrations of its active forms in blood. Under basal conditions (without additional niacin), active forms of niacin that already exist in the blood do not show significant correlations with metabolic parameters.

## 1. Introduction

Nicotinic acid and nicotinamide are two forms of water-soluble vitamin niacin. Both are precursors of the synthesis of nicotinamide adenine dinucleotide (NAD) and nicotinamide adenine dinucleotide phosphate (NADP). NAD and NADP are active forms of niacin and actively involved in many essential redox reactions in cellular metabolism, in the metabolism of lipids and carbohydrates, whereas NAD protects the organism from oxidative stress [1]. Although various niacin forms are nutritionally equivalent and can be used for the synthesis of NAD, their biological proportions vary and only nicotinamide can act as a reactive component [2].

For a long time, it was considered that the production of B vitamins in the rumen completely satisfies the needs of cows for niacin. The source of niacin can be ingested feed, or the animal itself can synthesize niacin through the enzymatic conversion of tryptophan and quinolone acid to niacin. Microorganisms in the rumen also synthesize niacin by using aspartate and dihydroxyacetone phosphate [3]. Firstly, research has shown that dairy cows do not need to be exogenously supplemented with B vitamin supplements, because the B vitamins in the diet and those synthesized by the ruminal microflora are sufficient in quantity. These attitudes have changed, especially due to metabolic changes in the calving period [4,5]. In this period, cows experience changes in the energy metabolism, metabolic stress, negative energy balance, insulin resistance, changes in lipid metabolism, reduced food intake and increased milk production [6,7], which directly or indirectly affect the production and availability of vitamins in the bodies of cows.

Vitamin niacin is of great importance for energy metabolism. Physiologically, niacin is incorporated into the coenzymes NAD and NADP [8]. They are involved in many metabolic processes: (a) in anabolic pathways (NADPH/NADP), such as the lipid synthesis of nucleic acids, NADPH is necessary as a reducing agent and (b) in catabolic pathways (NADH/NAD), NAD participates in a large number of oxidative–reductive reactions. In addition, NAD is a source of adenine dinucleotide phosphate (ADP)-ribose for protein modification. It is a precursor to two secondary nuclei (cADP-ribose and nicotinic acid adenine dinucleotide phosphate), which stimulate an increase in intracellular calcium concentration. Nicotinic acid in the mucosa of the small intestine is rapidly converted to NAD and then, by NAD+ glucohydrolase, excess NAD is hydrolyzed to nicotinamide, which is the main form of transport of niacin in the blood. NAD+ glucohydrolase is an enzyme that catalyzes the hydrolysis of NAD, producing ADP-ribose and nicotinamide. Nicotinamide is the primary form of niacin in circulation and is converted in the form of coenzyme (NAD and NADP) in tissue [3]. Niacin transport is associated with circulating erythrocytes, whereby it quickly leaves the bloodstream and enters the kidneys, liver or adipose tissue. Given the major discrepancies for the existence of nicotinic acid in circulation, nicotinamide is thought to be the main form of niacin transport in the blood, but nicotinic acid, which is not metabolized in the liver, is thought to be transported to various tissues in the body [9].

The biological effects of niacin in dairy cows are reflected in reduced lipolysis, ketogenesis, lipid accumulation in the liver and reduced insulin resistance, while the effect on glycemia is such that it helps maintain or increase glycemia [10,11,12,13,14]. This vitamin is important in milk production because it can also affect milk quality depending on the energy balance of cows and the stage of lactation, and it also shows positive effects on growth [15,16]. Therefore, this vitamin is used to treat negative energy balance or ketosis, especially in the peripartum period in cows [17]. The addition of niacin in ruminants affects the difference in the values of their vitamins, but in some experiments this effect was absent [18,19,20,21,22,23]. In some trials, the supplementation of nicotinic acid (NA) had positive effects on NA and nicotinamide (NAM) in the organism, while feeding with NAM led to a decrease in NA and NAM [23]. Tienken et al. [24] found that dietary NA increased serum nicotinamide concentrations, whereas NA could not be detected. Supplementation in sheep did not affect blood vitamin levels, but the injection of niacin into the duodenum resulted in its increase in the blood [25]. It is known that the biodegradability of niacin in the rumen is very high and that it takes 6–12 g of niacin to reach the duodenum to provide a biological effect on various blood parameters, such as a decrease in NEFA or BHB or an increase in glucose concentration in early lactation [26]. Therefore, rumen-protected niacin was made, which showed its effects on all aspects of metabolism, dry matter intake and milk production [27]. Excess niacin is thought to be rapidly converted to NAD which goes to the liver or adipocytes, and a disorder of its metabolism leads to various health disorders, such as fatty liver, low grade inflammation or insulin resistance [28]. There is a limited body of information on the NAD, NADP and niacin status response to niacin administration in dairy cows. One of the methods for assessing niacin status in humans is the measurement of the active coenzymes of niacin, NAD and NADP. Erythrocyte NAD and NADP concentration are more direct measures of functional niacin status and have been shown to respond to changes in niacin intake in humans [29,30].

In previous research [10,11,12,13,14,15,16,26,28], the effect of niacin has been observed through its biological effects on metabolic parameters, productivity and food intake in cows. So far, no association has been established between the niacin active form and metabolic parameters, whose values change under the influence of niacin. The aim of this study was to determine the concentration of NAD, NADP and their ratio in erythrocytes after niacin administration, as well as their relationship with the values of metabolic parameters in cows, to determine whether these niacin status indicators can be used to assess the biological effect of niacin in cow metabolic status.

## 2. Materials and Methods

### 2.1. Animals and Study Protocol

For this study, 90 clinically healthy Holstein-Friesian cows were chosen. The cows were in second and third lactation (3 to 4 years old) without a history of abortion and with milk production in previous lactation from 7650–8200 L. On the clinical examination, there were no signs of illness. Cows were divided into two groups: (1) Niacin group—45 healthy cows treated with nicotinic acid for 2 weeks prior to the expected partus, and 2 weeks after parturition; (2) Control group—45 healthy cows to whom niacin was not given. The experiment was established from February till May in a thermoneutral period of the year. Niacin in form of nicotinic acid was applied per os with food, 120 g per day per cow. Nicotinic acid was in the rumen-unprotected form (Rovimix^®^Niacin, Hoffmann-La Roche, AG, Switzerland). The dose of niacin was determined according to the following characteristic: the bioavailability of unprotected nicotinic acid is about 3–10% [31]; in most experimental models 3, 6 or 12 g/d (2.5–10% of 120 g/d) of protected nicotinic acid was supplemented to cows. The control group was not treated with nicotinic acid. The control group received a placebo of corn grain only. The diet for early lactation cows consisted of 3.43 kg alfalfa hay, 9.50 kg corn silage (44% DM), 9.0 kg corn silage (33.94% DM), 5.0 kg alfalfa haylage (47.40% DM), 5.0 kg brewer’s grain (21.0% DM), 2.50 kg corn grain, 1.50 kg barley grain, 1.30 kg soybean grits, 1.13 kg soybean meal (44% N), 1.30 kg wheat flour and 1.82 kg sugar beet pulp. Meal contained other components (total 1% DM): livestock salt; baking soda, 15-MgO, Premix, Phosphozel, Zenural (urea); Bentonite (Mycotoxin adsorbent), 20-Dairyfat c 16. Nutrient content of rations for experimental dairy cows in early lactation included: dry matter (DM) 21.5 kg; net energy of lactation 153.2 MJ/kg DM; crude protein (CP) 18.3% DM; rumen undegradable protein 39.69% CP; fat 4.92% DM; fiber 17.2% DM; acid detergent fiber (ADF) 22.6% DM; neutral detergent fiber (NDF) 37.16% DM. During the dry period the cows were fed a diet consisting of 3 kg lucerne hay, 3 kg wheat straw, 10 kg maize silage (30% DM), 4 kg lucerne haylage, 2 kg maize ear silage (68% DM), 0.5 kg dry sugerbeet pulp, 1.5 kg concentrate (12% CP). Nutrient content in daily rations for experimental dairy cows in the dry period: dry matter (DM) 12.10 kg; net energy of lactation (NEL), 66.2 MJ/kg DM; crude protein (CP) 12.10% DM; rumen undegradable protein (RUP) 35.82% CP; Fat 3.09% DM; Fibre 25.12% DM; acid detergent fibre (ADF) 32.33% DM; neutral detergent fibre (NDF) 49.08%DM and Iodine (I) 0.60 mg/kg (DM). The diets were in accordance to NRC recommendation [32].

### 2.2. Blood Collection and Analysis

Blood samples were collected a day before the experiment and a day after experiment for the determination of NAD, NADP and NAD:NADP or only after the experiment for the determination of blood biochemistry and endocrine parameters. Blood samples were collected before morning feeding by puncture of the coccygeal vein, using both lithium heparin 8-mL tubes and tubes with clot activator (BD Vacutainer tubes^®^ PST, BD, Switzerland), and were appropriately marked. The samples were kept on dry ice and protected from light until laboratory analysis. Samples were analyzed immediately after sampling. Parameters such as NAD, NADP and the NAD:NADP ratio was determined before and after experiment. NAD and NADP were determined using the colorimetric ELISA method from erythrocytes lysates. For the present study, kits from manufacturers Abcam were used and results were read from Rt-2100c ELISA microplate reader (Rayto, Shenzhen, China). The NAD:NADP ratio was calculated with a simple division of two numbers. Blood metabolites, such as glucose (GLU), nonesterified fatty acid (NEFA), beta-hydroxybutyrate (BHB), triglycerides (TGC), cholesterol (CHOL), total bilirubin (TBIL), aspartate aminotransferase (AST), gamma-glutamiltransferase (GGT), total protein (TPROT), albumin (ALB) and urea, were determined in the blood serum by standard colorimetric assay (Biosystems, Sp, Randox, UK) on a Chemray spectrophotometer (Rayto, Shenzen, China). Insulin concentration was measured by TOSOH AIA360 (Jpn). RQUICKIBHB and LFI index was calculated by formula from a previous study (Petrović et al., 2022) [33].

### 2.3. Statistical Analyses

The effects of niacin application on NAD, NADP and NAD:NADP ratio values were analyzed by using ANOVA analysis through the comparison of four subgroups: niacin group before treatment, niacin group after treatment, control group before treatment and control group after treatment. Effects of varying pretreatment baseline values of NAD, NADP and NAD:NADP on post-treatment measurements of the same variable were determined by ANCOVA analysis. Correlation between the active form of niacin and its ratio was determined by Pearson’s correlation coefficient and simple linear regression model, separately for experimental and control group. The ability to distinguish cows that received and did not receive additional niacin based on NAD, NADP and NAD:NADP ratio was examined by logistic regression and the determination of the area under the ROC curve (AUCROC). Relationships between NAD, NADP and NAD:NADP as independent factors with blood parameters as dependent factors were analyzed by Pearson’s correlation coefficient and simple linear regression model. The results are presented as linear regression equations and coefficients of determination and the regression curve and variability of parameters are presented graphically. Results are presented separately for the experimental and control groups. Statistical software SPSS 20.0 (IBM) was used for these purposes.

## 3. Results

The concentration of NAD and NADP was higher in the experimental group of cows receiving niacin, while the NAD:NADP ratio did not differ between the experimental and control groups of cows. The concentration of NAD, NADP and NAD:NADP ratio was not different in pretreatment period (Table 1). Cows in the experimental group can be clearly classified from cows in control group based on the area under the ROC curve of NAD (AUCROC = 0.937 ± 0.03; *p* ˂ 0.01), NADP (AUCROC = 0.948 ± 0.028; *p* ˂ 0.01) and NAD:NADP (AUCROC = 0.742 ± 0.052; *p* ˂ 0.01) (Figure 1). NAD and NADP show a positive correlation in the experimental and control groups (niacin group r^2^ = 0.33, *p* ˂ 0.01; control group r^2^ = 0.12; ˂0.01) (Figure 2). Changes in the NAD:NADP ratio in the function of NAD and NADP are also presented on same figure. Pretreatment baselines of NAD, NADP and the NAD:NADP ratio did not show significant effects on the post-treatment concentration of the same parameters (the results of ANCOVA analysis are not shown in the table or figure).

NAD, NADP and the NAD:NADP ratio correlate with metabolic profile parameters, and the strength and direction of the correlation depend on whether the cows received additional niacin through feed. NAD was positively correlated with glucose, insulin, triglyceride and LFI values, while it negatively correlated with BHB and urea levels in niacin-treated cows. In the control group, NAD negatively correlated with LFI and albumin values (Table 2, Figure 3).

NADP negatively correlates with insulin resistance indices, the glucose to insulin ratio (GLU:INS) and RQUICKI and positively with TBIL, GGT, ALB and LFI values in cows receiving niacin, while in the control group cows showed a negative correlation with GGT and a positive correlation with urea (Table 3, Figure 4).

The NAD:NADP ratio showed a positive correlation with glucose, insulin, GLU:INS and the RQUICKI index of insulin resistance, triglycerides and cholesterol and a negative correlation with NEFA, BHB, GGT and urea in cows receiving niacin, while in the control group cows showed a positive correlation with insulin and a negative correlation with albumin and urea (Table 4, Figure 5). The comparative analysis of coefficients of determination in statistically significant correlations suggests that NAD and NADP participate in the variability of tested metabolites in the range of 10 to 25%, while the NAD:NADP ratio participates in the variability of tested metabolites up to 56%.

## 4. Discussion

In the present study, very high niacin concentrations were fed to cows to elucidate niacin and metabolic pharmacokinetics for ruminants. There is a difference in the metabolism of nicotinic acid and nicotinamide, two forms of niacin, which are used for the biosynthesis of NAD [34]. There is no difference in the total amount of niacin in the rumen when comparing the ratio of concentrate and forage, when it ranges from 40:60 to 60:40, but a difference in the concentration of each vitamin individually was observed [35]. However, niacin synthesis improves with an increased amount of carbohydrates from non-fibrous foods, while the relationship between concentrate and foraged foods has no effect [36]. In humans, high dietary protein intake is associated with low NAD+ values in blood [37]. The microorganisms in rumen also synthesize niacin, whereby daily ruminal synthesis of niacin is greater than 2.2 g [38]. More niacin is synthesized from the microflora when less niacin is given as a supplement and vice versa [39]. The results of one study [35] showed that both forms, nicotinic acid and nicotinamide, are synthesized in significant amounts in the rumen. Most of the nicotinic acid and whole nicotinamide is therefore found in the bacterial cell, and there is a lack of absorption in the rumen because vitamins are present in the bacterial fraction. These forms of niacin are not available as free in the rumen content, which prevents their absorption. Erickson et al. [40] noted that the absorption of niacin through the rumen is possible but is limited because a small percentage (3–7%) of niacin is found as a supernatant ruminant fluid, and the largest amount of niacin is bound to cell microflora. The concentration of nicotinic acid in the duodenum is higher in cows fed with nicotinamide supplements (12 g daily) compared to cows fed nicotinic acid supplements [23]. When a higher dose of niacin is applied, more niacin is available to microorganisms than they need, so excess niacin can cover the lower part of the digestive tract. The concentration of niacin in the duodenum was higher in those animals in which niacin was administered [23,41]. Niacin is absorbed in the small intestine, whereby the absorption of niacin into the duodenum is not affected by diet [42]. Intestinal absorption of nicotinic acid is at a level of 73% and nicotinamide at a level of 94%, with an average of 84% of the total niacin in the duodenum [38].

Plasma NEFA concentrations were reduced from 546 µEq/L to 208 µEq/L one h after the abomasal infusion of 6 mg nicotinic acid per kg/tm and less than 100 µEq/L 3 h after the abomasal infusion of the two highest doses of nicotinic acid, which confirms the dose-dependent effect of niacin. Erickson et al. [43] obtained results regarding the significant effects of niacin on BHB, where the level of plasma BHB concentration was lower in niacin-fed cows compared to the control group. A significant reduction (*p* < 0.01) in BHB concentration was obtained in niacin-fed cows (in crystalline powder form) of 12 g per cow per day and a smaller reduction in cows given 6 g per cow per day compared to controls. NAD, NADP and the NAD:NADP ratio were correlated with metabolites consistently with the known biological effects of niacin. The application of niacin led to an increase in the values of these vitamins in the body, and with their increase, there is a decrease in NEFA, BHB, and urea and an increase in glucose, cholesterol, triglycerides, proteins and insulin sensitivity. Only an increased concentration of nicotinic acid in the blood can cause a decrease in the level of phosphorylation of hormone-sensitive lipase and prevent elevated levels of lipolysis and thus reduce the level of plasma NEFA [44,45]. The application of nicotinic acid, but not nicotinamide, can have positive effects on cows in the transition period. Since the accumulation of hepatic triglycerides is directly related to the concentration of NEFA in the blood, by reducing the concentration of NEFA in cows fed niacin in postpartum, it is possible to lead to a reduced accumulation of triglycerides and reduced problems with fatty liver [46]. Studies have shown that niacin supplements reduce the concentration of BHBA and NEFA in blood plasma with increasing serum glucose levels [47]. Insulin resistance in cows in the peripartum period occurs due to the priority use of glucose for fetal growth, udder development and lactation. Insulin resistance in Holstein cows has also been associated with increased plasma NEFA concentrations [48]. Niacin reduces lipolysis and stimulates an increase in glucose levels, which results in stimulation of insulin secretion and reduced insulin resistance. Data reported by Thornton and Schultz [49] showed altered glucose metabolism during the administration of a pharmacological dose of nicotinic acid in ruminants, i.e., increased plasma glucose and insulin concentrations as well as reduced glucose tolerance and insulin resistance. Such an increase may induce enhanced gluconeogenetic activity at the cellular level promoted by nicotinic acid-induced partial suppression of lipogenesis. Pires et al. [48] suggested that lower concentrations of NEFA, achieved with nicotinic acid administration in Holstein cattle on a restrictive diet, improve insulin response and glucose utilization with increased insulin sensitivity, implying that NEFA in the blood is an important factor in the development of insulin resistance in dairy cows during a negative energy balance. Niacin significantly affects glucose concentration, whereby an increase in glucose concentration was dependent on niacin concentration and length of treatment. Pescara et al. [50] stated that the mechanism by which nicotinic acid increases plasma glucose concentration is unclear and this may be the result of increased hepatic glucose production, decreased blood glucose clearance, or both. Titgemeyer et al. [51] stated that it is difficult to see whether an increase in glucose leads to an increase in insulin or insulin resistance leads to an increase in glucose.

Ghorbani et al. [52] noted that total plasma protein concentrations in niacin and negative control groups of cows increased during the first 4 week postpartum and then tended to decrease until 8th week postpartum. A significant increase in protozoan populations was observed during the feeding of cows with unprotected niacin [53,54,55]. The increase in protozoa has an effect on the metabolism of nitrogen in the rumen and results in increased synthesis of microbial proteins [55]. Favoured protein synthesis in niacin supplementation may be since ammonia fixation in the ruminal microflora is performed via NADP or NAD-linked glutamyl dehydrogenase, and niacin is a precursor to pyridine nucleotide synthesis [56]. This increased utilization of ammonia for protein synthesis in rumen reduces the amount of excess ammonia that needs to be transported through the rumen wall to the liver, where it is converted to urea.

In the present study, correlations between active forms of niacin in blood and metabolites were realized only in the group where niacin is added. This suggests that niacin active forms in blood can be useful in assessing metabolic status due to exogenously added niacin, while its action in basal conditions cannot be assessed based on vitamin and metabolite links. According to literature data, a two-week treatment of mice with high doses of nicotinic acid and nicotinamide (500 and 1000 mg/kg) has an effect on NAD levels in various tissues, so an increased concentration of NAD by 40 to 60% in blood and liver was found [57]. This indicates the ability of niacin to stimulate NAD synthesis in the liver and blood and that nicotinamide can be converted to an alternative form. Thus, the bioavailability of a nicotinamide increase and/or nicotinamide treatment can induce cellular adaptation, leading to better NAD biosynthesis [58]. Jackson et al. [57] showed that nicotinic acid can increase the concentration of NAD in the liver and blood, similar to nicotinamide. Hara et al. [59] indicated that exogenous added nicotinic acid induces a significant increase in cellular NAD levels in human cells, while nicotinamide added at the same concentration does not cause a significant increase in NAD concentration. Based on that, it was concluded that nicotinic acid is a better substrate for increasing cellular levels of NAD than nicotinamide. Erythrocyte concentrations of NAD and NADP are a direct indicator of the functional status of niacin since they are more responsive to changes in niacin intake, so the NAD:NADP ratio may be a useful indicator for measuring niacin status in humans [29]. In people who received 50% less than the recommended daily dose of niacin, a 70% reduction in NAD concentrations was observed within 5 weeks, while the NADP concentration remained relatively constant [29]. In our studies, the NAD:NADP ratio did not show a statistically significant difference between the niacin group and the control group. Almost all cows in the experimental group that received niacin had a NAD:NADP ratio higher than 2. Under normal conditions, the concentration of erythrocyte NAD is higher than the concentration of NADP and therefore the NAD:NADP ratio should always be >1 [60]. It is known that the erythrocyte NADP concentration is usually constant and any reduction in erythrocyte NAD will reduce the NAD:NADP ratio and vice versa. Based on that, there is a suggestion that any niacin index below 1.0 may be an indication of niacin deficiency in humans [29]. Due to all the above, the measurement of these vitamins and linking their concentration with the concentration of selected metabolites is a good indicator of the biological effect of niacin. The latest review has shown a link between NAD values, i.e., their forms of NAD+ and NADH in ageing and the protection against apoptosis, carcinogenesis, neurodegenerative and metabolic diseases [61]. It is believed that the cell first produces NAD. After that, with the help of the enzyme NAD-kinase, phosphate is added to the molecule when NADP is formed [62], which is important for anabolic pathways in the cell.

In vivo studies demonstrated that the application of pharmacological doses of nicotinic acid reduces the level of plasma NEFA by inhibiting lipolysis in cattle [51]. This antilipolytic potential of nicotinic acid is most likely realized through action on the niacin receptor GPR109A [63]. The GPR109A antilipolytic pathway, already described in other animal species, has recently been shown to exist also in a functional form in bovine tissue in vitro [44]. On the other hand, nicotinamide has a very low affinity for GPR109A. The activation of GPR109A with nicotinic acid leads to the inhibition of adrenalin cyclase activity, leading to a decrease in cAMP concentration in cells. Decreased cAMP in adipocytes leads to a consequent inactivation of protein kinase A and a decrease in phosphorylation of hormone-sensitive lipase and, thus, to a reduction in lipolysis [63]. The GPR109A receptor is present in adipose tissue and immune cells, and, in cattle, it has also been found in muscle, brain and liver [64]. In cattle, the GPR109A ligand, nicotinic acid, nicotinamide and BHB showed different levels of efficacy in induced antilipolysis under in vitro conditions. Nicotinic acid reduces the phosphorylation of hormone-sensitive lipase and thus reduces the lipolytic response, whereby nicotinamide is not able to suppress lipolytic activity in bovine tissue in vitro. BHB only at its highest concentration induces a significant reduction in glycerol release and phosphorylation of hormone-sensitive enzyme [44]. NAD+ in adipose tissue has a significant effect on the balance of sirtuin and PARP, and metabolic pathways differ in obese and normally fed individuals [65]. Anti-inflammatory drugs via NADPH oxidase affect the inhibition of lipolysis [66]. In humans and rodents, the liver is the main organ for the de novo synthesis of fatty acids from glucose, while in ruminants, adipose tissue is the main site of lipogenetic activity [9]. In these animals, acetate is the primary substrate for the de novo synthesis of fatty acids in adipose tissue. De novo fatty acid synthesis requires an adequate supply of cytosols with NADPH, whereby fourteen molecules of acetyl CoA and NADPH are required for this synthesis of one molecule of palmitic acid. Pentose phosphate cycle activities are thought to be major sources of NADPH for lipogenesis. Baldwin et al. [67] found that in lactating cows, 64% of the NADPH required for lipogenesis in adipose tissue was generated from the pentose phosphate cycle. In the process of lipogenesis, NADPH is an agent that performs reduction, i.e., donates an electron, and NADP+ is obtained. Another important metabolic pathway in which NAD participates is β-oxidation of fatty acids in liver mitochondria [54]. Therefore, niacin vitamins have links with various metabolites that represent the metabolism of fats and carbohydrates in the periparturient period.

## 5. Conclusions

In conclusion, the application of niacin in cows leads to an increase in NAD and NADP in the blood. The NAD:NADP ratio was not significantly altered by the effect of niacin administration, NAD and NADP are good indicators of the ability of additional niacin source to create functional cofactors due to their concentration change, while the NAD:NADP ratio is a good indicator of the biological effects of additional niacin due to correlation with many metabolites. Under basal conditions (without additional niacin), active forms of niacin that already exist in the blood do not show significant correlations with metabolic parameters.

## Figures and Tables

**Figure 1 animals-12-01524-f001:**
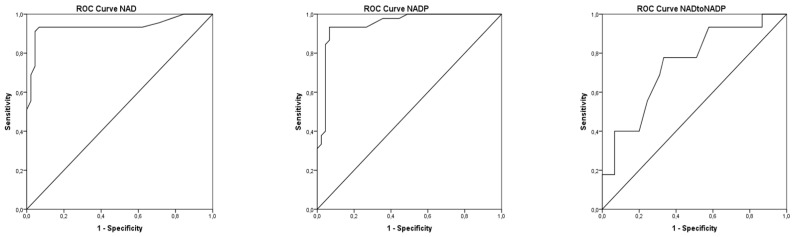
ROC curve for niacin status indicators NAD, NADP and NAD:NADP for distinguishing cows that received and did not receive additional niacin in food.

**Figure 2 animals-12-01524-f002:**
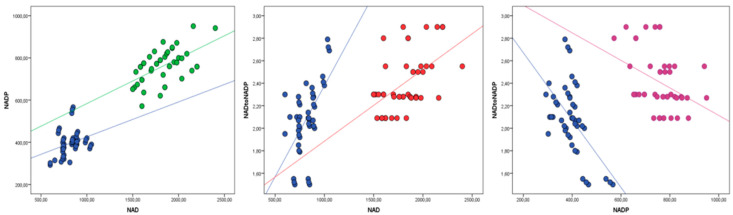
Linear regression curves between niacin status indicators NAD, NADP and NAD:NADP in cows that received (green, red, purple) and did not receive niacin (blue).

**Figure 3 animals-12-01524-f003:**
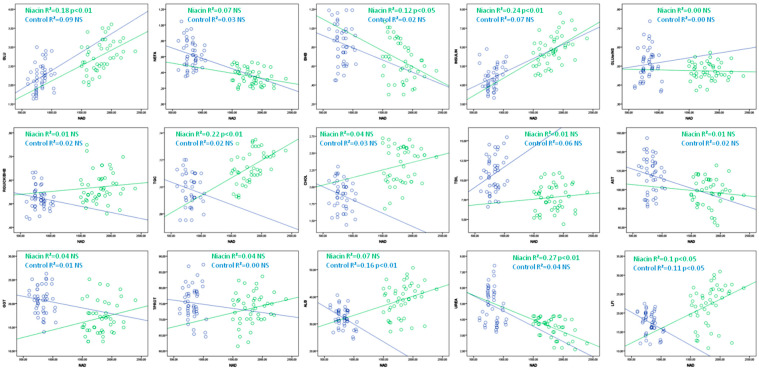
Linear regression line between NAD and metabolic parameters in cows receiving (green) and not receiving niacin (blue).

**Figure 4 animals-12-01524-f004:**
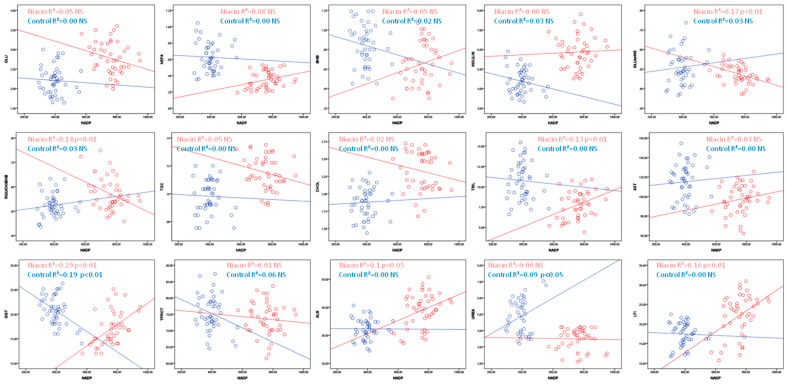
Linear regression line between NADP and metabolic parameters in cows receiving niacin (red) and not receiving niacin (blue).

**Figure 5 animals-12-01524-f005:**
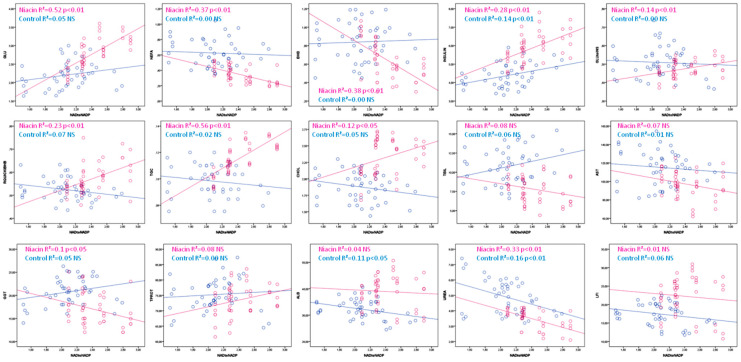
Linear regression line between the NAD:NADP index and metabolic parameters in cows receiving (purple) and not receiving niacin (blue).

**Table 1 animals-12-01524-t001:** Influence of niacin application on NAD, NADP and the NAD:NADP ratio. Abbreviations: Nicotinamide adenine dinucleotide (NAD), nicotinamide adenine dinucleotide phosphate (NADP), ratio (NAD:NADP).

Active Form of Niacin	Niacin		Control		*p*
	Pretreatment	After Treatment	Pretreatment	After Treatment	
NAD (nmol/L)	809.1 ± 210.1 ^a^	1761.9 ± 344.8 ^b^	815.8 ± 199.1 ^a^	863.6 ± 217.8 ^a^	˂0.01
NADP (nmol/L)	399.4 ± 106.4 ^a^	737.8 ± 118.2 ^b^	400.3 ± 99.6 ^a^	412.5 ± 101.8 ^a^	˂0.01
NAD:NADP	2.02 ± 0.35 ^a^	2.38 ± 0.29 ^a^	2.03 ± 0.33 ^a^	2.11 ± 0.31 ^a^	NS

^a,b^—different superscript means statistic significant difference between niacin and control groups.

**Table 2 animals-12-01524-t002:** Relationship of metabolic parameters with NAD values in cows that received and did not receive additional niacin. Abbreviations: Nicotinamide adenine dinucleotide (NAD), glucose (GLU), nonesterified fatty acid (NEFA), beta-hydroxybutyrate (BHB), glucose to insulin ratio (GLU:INS), Revised Quantitative Insulin Sensitivity Check Index with BHB (RQUICKIBHB), triglycerides (TGC), cholesterol (CHOL), total bilirubin (TBIL), aspartate aminotransferase (AST), gamma-glutamiltransferase (GGT), total protein (TPROT), albumin (ALB) and liver functionality index (LFI).

	Niacin	R^2^	*p*	Control	R^2^	*p*
GLU	=1.3 + 0.0008 × NAD	0.18	˂0.01	=1.41 + 0.005 × NAD	0.09	NS
NEFA	=0.58 − 0.00013 × NAD	0.07	NS	=0.84 − 0.00061 × NAD	0.03	NS
BHB	=1.26 − 0.0003 × NAD	0.12	˂0.05	=1.05 − 0.00026 × NAD	0.02	NS
INSULIN	=2.52 + 0.0018 × NAD	0.24	˂0.01	=3.16 + 0.0015 × NAD	0.07	NS
GLU:INS	=0.49 − 0.00006 × NAD	0.00	NS	=0.47 + 0.00005 × NAD	0.00	NS
RQUICKIBHB	=0.53 + 0.00002 × NAD	0.01	NS	=0.57 − 0.00005 × NAD	0.02	NS
TGC	=0.07 + 0.00003 × NAD	0.22	˂0.01	=0.11 − 0.000017 × NAD	0.02	NS
CHOL	=1.91 + 0.00022 × NAD	0.04	NS	=2.2 − 0.0004 × NAD	0.03	NS
TBIL	=6.49 + 0.00073 × NAD	0.01	NS	=6.51 + 0.0052 × NAD	0.06	NS
AST	=108 − 0.006 × NAD	0.01	NS	=132 − 0.02 × NAD	0.02	NS
GGT	=11.28 + 0.0032 × NAD	0.04	NS	=22.74 − 0.0024 × NAD	0.01	NS
TPROT	=65.4 + 0.0045 × NAD	0.04	NS	=77.4 − 0.0022 × NAD	0.00	NS
ALB	=26.09 + 0.0071 × NAD	0.07	NS	=42.91 − 0.01 × NAD	0.16	˂0.01
UREA	=6.3 − 0.0016 × NAD	0.27	˂0.01	=6.46 – 0.0019 × NAD	0.04	NS
LFI	=8.28 + 0.0077 × NAD	0.1	˂0.05	=24.53 − 0.0087 × NAD	0.11	˂0.05

NS—non significant, *p* > 0.05.

**Table 3 animals-12-01524-t003:** Relationship of metabolic parameters with NADP values in cows that received and did not receive additional niacin. Abbreviations: nicotinamide adenine dinucleotide phosphate (NADP), glucose (GLU), nonesterified fatty acid (NEFA), beta-hydroxybutyrate (BHB), glucose to insulin ratio (GLU:INS), Revised Quantitative Insulin Sensitivity Check Index with BHB (RQUICKIBHB), triglycerides (TGC), cholesterol (CHOL), total bilirubin (TBIL), aspartate aminotransferase (AST), gamma-glutamiltransferase (GGT), total protein (TPROT), albumin (ALB) and liver functionality index (LFI).

	NIACIN	R^2^	*p*	CONTROL	R^2^	*p*
GLU	=3.69 − 0.0012 × NADP	0.05	NS	=2.32 − 0.000265 × NADP	0.00	NS
NEFA	=0.06 + 0.00038 × NADP	0.08	NS	=0.67 − 0.00016 × NADP	0.00	NS
BHB	=0.22 + 0.00057 × NADP	0.05	NS	=1.03 − 0.00047 × NADP	0.02	NS
INSULIN	=5.57 + 0.00039 × NADP	0.00	NS	=5.15 − 0.0019 × NADP	0.03	NS
GLU:INS	=0.65 − 0.00024 × NADP	0.17	˂0.01	=0.47 + 0.00011 × NADP	0.01	NS
RQUICKIBHB	=0.8 − 0.00031 × NADP	0.14	˂0.01	=0.49 + 0.00009 × NADP	0.01	NS
TGC	=0.14 − 0.000032 × NADP	0.05	NS	=0.1 − 0.000059 × NADP	0.00	NS
CHOL	=2.75 − 0.00057 × NADP	0.02	NS	=1.82 + 0.00034 × NADP	0.00	NS
TBIL	=2.09 + 0.0075 × NADP	0.13	˂0.05	=11.53 − 0.002 × NADP	0.00	NS
AST	=73.48 + 0.03 × NADP	0.03	NS	=109 + 0.02 × NADP	0.00	NS
GGT	=0.4 + 0.02 × NADP	0.29	˂0.01	=29.02 − 0.02 × NADP	0.19	˂0.01
TPROT	=76.99 − 0.0046 × NADP	0.01	NS	=83.81 − 0.02 × NADP	0.06	NS
ALB	=21.28 + 0.02 × NADP	0.1	˂0.05	=32.42 − 0.0004 × NADP	0.00	NS
UREA	=3.61 − 0.0002 × NADP	0.00	NS	=2.72 + 0.0054 × NADP	0.09	˂0.05
LFI	=2.14 + 0.03 × NADP	0.16	˂0.01	=18.1 − 0.0017 × NADP	0.00	NS

NS—non significant, *p* > 0.05.

**Table 4 animals-12-01524-t004:** Relationship of metabolic parameters with NAD:NADP values in cows that received and did not receive additional niacin. Abbreviations: Nicotinamide adenine dinucleotide to nicotinamide adenine dinucleotide phosphate ratio (NAD:NADP or NADtoNADP), glucose (GLU), nonesterified fatty acid (NEFA), beta-hydroxybutyrate (BHB), glucose to insulin ratio (GLU:INS), Revised Quantitative Insulin Sensitivity Check Index with BHB (RQUICKIBHB), triglycerides (TGC), cholesterol (CHOL), total bilirubin (TBIL), aspartate aminotransferase (AST), gamma-glutamiltransferase (GGT), total protein (TPROT), albumin (ALB) and liver functionality index (LFI).

	Niacin	R^2^	*p*	Control	R^2^	*p*
GLU	=−0.02 + 1.16 × NADtoNADP	0.52	˂0.01	=1.65 + 0.27 × NADtoNADP	0.05	NS
NEFA	=0.94 − 0.25 × NADtoNADP	0.37	˂0.01	=0.7 − 0.03 × NADtoNADP	0.00	NS
BHB	=1.86 − 0.5 × NADtoNADP	0.38	˂0.01	=0.78 + 0.03 × NADtoNADP	0.00	NS
INSULIN	=1.91 + 1.7 × NADtoNADP	0.28	˂0.01	=2.8 + 0.77 × NADtoNADP	0.14	˂0.01
GLU:INS	=0.3 + 0.07 × NADtoNADP	0.14	˂0.01	=0.54 − 0.02 × NADtoNADP	0.00	NS
RQUICKIBHB	=0.27 + 0.12 × NADtoNADP	0.23	˂0.01	=0.6 − 0.04 × NADtoNADP	0.07	NS
TGC	=0.03 + 0.03 × NADtoNADP	0.56	˂0.01	=0.11 − 0.005 × NADtoNADP	0.02	NS
CHOL	=1.43 + 0.37 × NADtoNADP	0.12	˂0.05	=2.21 − 0.16 × NADtoNADP	0.05	NS
TBIL	=11.8 − 1.7 × NADtoNADP	0.08	NS	=6.26 + 2.15 × NADtoNADP	0.06	NS
AST	=133 − 14.9 × NADtoNADP	0.07	NS	=128 − 6.18 × NADtoNADP	0.01	NS
GGT	=27.1 − 4.2 × NADtoNADP	0.1	˂0.05	=15.6 + 2.5 × NADtoNADP	0.07	NS
TPROT	=60.4 + 5.45 × NADtoNADP	0.08	NS	=72.34 + 1.4 × NADtoNADP	0.00	NS
ALB	=42.5 − 1.47 × NADtoNADP	0.04	NS	=40.4 − 3.9 × NADtoNADP	0.11	˂0.05
UREA	=6.93 − 1.44 × NADtoNADP	0.33	˂0.01	=7.86 − 1.43 × NADtoNADP	0.16	˂0.01
LFI	=26.7 − 1.9 × NADtoNADP	0.01	NS	=22.03 − 2.21 × NADtoNADP	0.06	NS

NS—non significant, *p* > 0.05.

## Data Availability

Not applicable.

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
