# Peer review of "Niacin Status Indicators and Their Relationship with Metabolic Parameters in Dairy Cows during Early Lactation"

_animals, 2022, doi:10.3390/ani12121524_

Round 1
Reviewer 1 Report
This article was conducted to study Niacin status indicators and their relationship with metabolic parameters in dairy cows during early lactation. This study provided a nutritional monitoring method that niacin status indicators can be used to assess the biological effect of niacin of cow metabolic status. But there are some questions that should be taken into account.
Here are some specific suggestions.
- Line 146, “the bioavailability of unprotected nicotinic acid is about 10%”, this sentence should list references.
- Line 156, Diet composition and nutrition content should be expressed in percentage, and net energy of lactation should be expressed in MJ/kg of DM. Because the feed intake of the cow may not be consistent with the diet given in this experiment.
Author Response
- Line 146, “the bioavailability of unprotected nicotinic acid is about 10%”, this sentence should list references. Thank you. It is estimated that only 3 to 10% of unprotected niacin escaped from ruminal degradation when feeding 2 to 12 g of unprotected niacin (Miller et al., 1986; Zinn et al., 1987; Santschi et al., 2005 cited in Rungruang, S.; Collier, J. L.; Rhoads, R. P.; Baumgard, L. H.; De Veth, M. J.; Collier, R. J. A dose-response evaluation of rumen-protected niacin in thermoneutral or heat-stressed lactating Holstein cows. J Dairy Sci 2014, 97, 5023-5034). The application of 120 g / d allows 3.6 to 12 g / d NA to be found in the rumen after metabolism, on average 7.8 g / d NA, which will go to the abomazum. It was found that the antilipolytic effect is best achieved if the cows in abomazum received 6 or 10 mg / kg BW, or on a conditional throat of 500 kg that is 3 to 5 g / day (Pires and Grumer, Journal of Dairy Science, Volume 90, Issue 8, August 2007, Pages 3725-3732).
- Line 156, Diet composition and nutrition content should be expressed in percentage, and net energy of lactation should be expressed in MJ/kg of DM. Because the feed intake of the cow may not be consistent with the diet given in this experiment. Thank you.
Reviewer 2 Report
This research studied the effects of niacin supplementation on concentrations of NAD,NADP, and NAD:NADP ratio, and relations between several metabolites and NAD,NADP, or NAD:NADP ratio were detected. The authors proposed that NAD, NADP, and NAD:NADP ratio are good indictors of the niacin's effects on metabolism. I'm wondering if the NA and NAM concentrations in the blood were also determined or not. Because the conclusion states that "NAD and NADP are good indicators of niacin absorption ", NA or NAM concentrations are direct indictors of niacin absorption, which are quite necessary in the current study. In addition, did the niacin supplementation influence the metabolites concentrations? Furthermore, the discussion is too broad and it should be more focused on the findings of the current study, e.g. why the NAD,NADP, or NAD:NADP ration was correlated with some metabolites in the niacin group, not in the control group, or vice versa? Did blood NA and those metabolites concentrations have the similar correlation?
Author Response
This research studied the effects of niacin supplementation on concentrations of NAD,NADP, and NAD:NADP ratio, and relations between several metabolites and NAD,NADP, or NAD:NADP ratio were detected. The authors proposed that NAD, NADP, and NAD:NADP ratio are good indictors of the niacin's effects on metabolism. I'm wondering if the NA and NAM concentrations in the blood were also determined or not. Because the conclusion states that "NAD and NADP are good indicators of niacin absorption ", NA or NAM concentrations are direct indictors of niacin absorption, which are quite necessary in the current study. In addition, did the niacin supplementation influence the metabolites concentrations? Furthermore, the discussion is too broad and it should be more focused on the findings of the current study, e.g. why the NAD,NADP, or NAD:NADP ration was correlated with some metabolites in the niacin group, not in the control group, or vice versa? Did blood NA and those metabolites concentrations have the similar correlation?
We agree that the sentence in the conclusion is awkwardly formulated, because NA and NAM are direct indicators of niacin absorption. Is it OK to write: " NAD and NADP are good indicators of the ability of additional niacin source to create functional cofactors due to their concentration change, while NAD: NADP ratio is good indicator of biological effects of additional niacin due to correlation with many metabolites". We did not measure NA and NAM in this experiment. The reason is that the focus here was on examining the biological effects of niacin. The first step in achieving a biological effect is the creation of functional cofactors derived from vitamin B3 are nicotinamide adenine dinucleotide (NAD +), its phosphorylated form, nicotinamide adenine dinucleotide phosphate (NADP +). Nicotinic acid and its nicotinamide derivative will be used as building block for the biosynthesis of the coenzymes nicotinamide adenine dinucleotide (NAD) and nicotinamide adenine dinucleotide phosphate (NADP). These compounds are used as indispensable cofactors by about 200 different classes of dehydrogenases. NAD functions mainly in catabolic reactions that generate energy by biological oxidation of carbohydrates, proteins, and fatty acids. NADP is involved mainly in anabolic reactions to build up cell mass. Also, it is known that the ability of NA to increase cellular NAD contents may account for some of the clinically observed effects of the vitamin. So far, I have not found results that describe the correlation of NA or NAM with metabolic parameters in cows. In general, longitudinal studies have been performed so far where the association of NA / NAM with metabolites can be sensed over time. However, we did not find examples of cross-sectional studies. Nicotinic acid has a half-life of about 60 minutes, because it immediately turns into NAD, which is an additional reason why we decided to determine the active form of niacin such as NAD / NADP.
Reviewer 3 Report
Specific comments for the authors
L18: represent
L24-25: language improvement
L38/143: orally or per os
L64: different/various niacin forms
L78 affect
L125 citation needed here
L134: the feeding details for dry period are not provided. Authors give detailed information about early lactation feeding, but the cows of the study were also under dry period feeding for the hald study period
L162 samples
L160: the description of blood sampling as before and after experiment is not precise. Definite description is needed, about what is before (the day before niacin addition, the day of addition?) and what is after (last day of additional niacin feeding? A time point after the cease of this fedding?)
Additional useful details are needed for the study design: were all 90 animals in the same time period? Which time of year? If the duration of experiment is long e.g. 6 months till all cows calve, possible heat stress in summertime would have impact in measurement and feed consumption.
L172: these blood metabolite measured should be named here with the abbbreviation used
L183: were
L189-191: repeated sentence
L193-94 ? this is not clear in meaning, needs rephrase.
L197-199 needs improvement in meaning
L200: pretreatment period as a term is misleading and not defined (as above mentioned), as period of time but more as a pretreatment sample
L287-288: needs revision in meaning
L324-325: needs revision in meaning
L331: Holstein
L332-333: needs revision in meaning
L3352-354: needs revision in meaning
Authors provide too many references, many of them are used once and always with others, so they can be omitted e.g. 27,48,57, 64-69, 72, 76.
Author Response
Thanks for all the specific and useful corrections. We have tried to correct everything according to your requirements.
Specific comments for the authors
L18: represent OK
L24-25: language improvement Before: We concluded NAD and NADP are excellent indicators that feeding niacin is readily available to the body, but the NAD:NADP ratio is a better indicator of niacin’s effect on affecting change of metabolic profile in dairy cows during early lactation.
After: We concluded that NAD and NADP are good indicators of the ability of additional niacin source to create functional cofactors due to their concentration change, while NAD: NADP ratio is good indicator of biological effects of additional niacin due to correlation with many metabolites
L38/143: orally or per os OK
L64: different/various niacin forms OK
L78 affect OK
L125 citation needed here – Agree. “In previous research [10-16, 26,29],….”
L134: the feeding details for dry period are not provided. Authors give detailed information about early lactation feeding, but the cows of the study were also under dry period feeding for the hald study period – Agree. We have now added a meal during the dry period.
L162 samples OK
L160: the description of blood sampling as before and after experiment is not precise. Definite description is needed, about what is before (the day before niacin addition, the day of addition?) and what is after (last day of additional niacin feeding? A time point after the cease of this fedding?)– “Blood samples were collected in day before experiment and day after experiment….”
Additional useful details are needed for the study design: were all 90 animals in the same time period? Which time of year? If the duration of experiment is long e.g. 6 months till all cows calve, possible heat stress in summertime would have impact in measurement and feed consumption. - Experiment was established from February till May in thermoneutral period of year.
L172: these blood metabolite measured should be named here with the abbbreviation used – “Blood metabolites such as glucose (GLU), nonesterified fatty acid (NEFA), beta-hydroxybutyrate (BHB), triglycerides (TGC), cholesterol (CHOL), total bilirubin (TBIL), aspartate aminotransferase (AST), gama glutamiltransferase (GGT), total protein (TPROT), albumin (ALB) and urea were determined in blood serum by….”
L183: were OK
L189-191: repeated sentence Agree. Sentence deleted.
L193-94 ? this is not clear in meaning, needs rephrase. Before: The paper will present linear regression equations and the coefficient of determination especially for the group that received and did not receive niacin. Now: The results will be presented as linear regression equations, coefficients of determination and graphically present of regression curve and variability of parameters, separately for the experimental and control groups.
L197-199 needs improvement in meaning - Before: In cows receiving niacin, there was a significantly higher concentration of NAD and NADP, whereas the value of NAD:NADP ratio did not differ in these two groups of cows. Now: The concentration of NAD and NADP was higher in the experimental group of cows receiving niacin, while the NAD: NADP ratio did not differ between the experimental and control groups of cows.
L200: pretreatment period as a term is misleading and not defined (as above mentioned), as period of time but more as a pretreatment sample – We have now defined in material and methods.
L287-288: needs revision in meaning Before: The contents of the rumen are not available in free form to be absorbed. Now: These forms of niacin are not available as free in the rumen content, which prevents their absorption.
L324-325: needs revision in meaning - This sentence is unnecessary and deleted, because the essence is stated in the sentence before and after it.
L331: Holstein OK
L332-333: needs revision in meaning – Before: Since niacin helps reduce lipolysis and raises glycemia, it can increase insulin secretion and efficiency and reducing insulin resistance. Now: Niacin reduces lipolysis and stimulates an increase in glucose levels, which results in stimulation of insulin secretion and reduced insulin resistance.
L352-354: needs revision in meaning Before: Ghorbani et al. [53] noted an increase in total plasma protein concentrations during the first 4 weeks of postpartum, and a tendency to decrease to 8 weeks of postpartum in the niacin-receiving and non-niacin groups. Now: Ghorbani et al. [53] noted that total plasma protein concentrations in niacin and negative control groups of cows increased during the first 4 week postpartum and then tended to decrease until 8th week postpartum.
Authors provide too many references, many of them are used once and always with others, so they can be omitted e.g. 27,48,57, 64-69, 72, 76. Agree. We have now deleted the unnecessary references.
Round 2
Reviewer 2 Report
L40:"per os with feed"
L49:"feed", and please check others as well
L194:“of for”?
Table1-4: Abbreviation and letters should be defined in the table caption.
Figures:R2 and p values should be shown in the figures as well.
L298: confusing sentence, please re-write it.
L321-334:It ’s not necessary to repeat the reason of niacin dosage used in the current study, it’s already in the introduction.
L333-4: improper statement here, please re-write it.
L389: Niacin is just one kind of vitamins, please carefully extrapolate the results, and please checks the others
L470-472: "Intensity of the biological effect of additional niacin can be accurately determined only if we know the concentrations of its active forms in blood." is not an appropriate conclusion.
Reference:The format of references should be kept uniform.
Author Response
We fully agree with all the reviewer's findings.
L40:"per os with feed" ОК
L49:"feed", and please check others as well ОК
L194:“of for”?OK - of four
Table1-4: Abbreviation and letters should be defined in the table caption. AGREE
Figures:R2 and p values should be shown in the figures as well. AGREE
L298: confusing sentence, please re-write it. OK
L321-334:It ’s not necessary to repeat the reason of niacin dosage used in the current study, it’s already in the introduction. AGREE sentence deleted
L333-4: improper statement here, please re-write it. The sentence was deleted, because in the last version of the manuscript, everything about the doses was explained in the material and methods with reference.
L389: Niacin is just one kind of vitamins, please carefully extrapolate the results, and please checks the others. AGREE
L470-472: "Intensity of the biological effect of additional niacin can be accurately determined only if we know the concentrations of its active forms in blood." is not an appropriate conclusion. AGREE. A very broad inductive conclusion, which we have deleted.
Reference:The format of references should be kept uniform. OK, Thank you.
This manuscript is a resubmission of an earlier submission. The following is a list of the peer review reports and author responses from that submission.
Round 1
Reviewer 1 Report
Line/Comment
28 …active forms of niacin…
22 …indicate the vitamin status in …
23 ratios for values
24 …receiving additional niacin.
25 …indicators that feeding niacin is readily available to the body.
26 …of niacin’s effect on affecting change of…
31 …with metabolic…
35 …but the ratio of…did not differ compared with control.
43/44 This seems confusing to myself. Can you restate?
49 delete “the”
50 Both are precursors for …
52 being for and
54 although niacin forms are…
104 What is yellow-protected niacin?
107 Such as?
General:
- I understand that English might not be the first language of the authors, but there is a lot of editing that can be done to make the manuscript more reader friendly. You see some of my suggestions above, but I don’t believe that I should have to correct this. I don’t want to be discouraging to the authors, but can someone help you shorten and strengthen the English? For example lines 244 to 245, In the present study, very high niacin concentrations were fed to elucidate niacin and metabolic pharmacokinetics for ruminants. The verbose writing can readily be revised to shorten the statements but still make the points.
- What is blush? Some words just seem out of place which I’m sure is a language translation within Word.
- You don’t need 77 references. Your study is in cows, stay focus on cows, unless this study has applications to humans, I suggest removing most of the human discussion. Stay focus on the cow.
- Do you have any incidences of ketosis in your study? Where is the production data? Published in another paper? What exactly is the objective of this report? Is is to feed higher niacin and determine what metabolic parameters change?
- Niacin is a vitamin, NAD and NADP are not and the ratio of NAD/NADP is not a metabolic. Also watch out for ration being used for ratio.
- What is sirtuin?
- FYI, I fed the 36 g/d niacin as a consulting nutritionist and it did not help in that situation.
- I think rumen protected niacin should be in the title. Rumen protected and unprotected niacin is a big difference. Then limit references to those from rumen protected niacin and only bring in key unprotected niacin references.
- Lastly my suggestion is to suggest using NAD/NADP as a means to reduce clinical and subclinical ketosis in postfresh dairy cows. This is where a table of clinical and subclinical ketosis characterizations of the treatments would really help the paper.
Reviewer 2 Report
This study was to determine the relationship between niacin active form- NAD, NADP and NAD: NADP ratio with values of metabolic parameters in cows. The main concerns are the sampling procedure. The authors only take one-time blood samples two weeks after calving. Then the question is how the authors control the individual background noise. The concertation before applying niacin should be used as a covariate for statistical analysis to correct the effect of niacin because some animals might genetically have higher NAD NADP concentrations than others. Also, since the animals received niacin two weeks before calving, could the author justify why blood samples were not taken two weeks before calving. And please provide how the two groups’ animals were divided, based on which criteria? And when the authors performed statistics, where these factors were considered in the model? If yes, please specify? The current paper would be more interesting if you take the blood samples before calving and then determine the change of NAD NADP concentration during calving, which would be more fitted to your objective. And please make sure the title and legacy of the figure are matched, and explain when you use abbreviations that the journal did not suggest. There are many tables and figures, but they provide the same information. Either one of them should go to supplementary materials to make the manuscript tidier.
Line 20-21: It is not clear about what is “metabolic changes in the function of niacin form and dose”, what kinds of metabolic changes, and what is the function of niacin form and does”, and in which species of animals?
Line 23: what do you mean by “different values”? Do you mean concentration? It is not apparent here. And what was the physiological status of cows?
Line 24: Where were the NAD and NADP measured (etc., blood)?
Simple summary: There are several unclear terminologies, in summary, making it hard to understand (metabolic profiles, functions, vitamin). It is hard to understand the main finding of the current study and its application.
Line 30-31: As you evaluate the cows at the early lactation stage, please provide this information in your aim
Line 31-32: please provide the source of niacin for cows and how cows received niacin.
Line 33-34: were blood samples also taken before receiving niacin and before parturition?
Line 34-35: please provide units of NAD and NADP.
Line 40-41: only the cows that received niacin were used for correlation analysis? If you did not also perform for the control group, how you could conclude “NAD: NADP is a good indicator of the biological effect of applied niacin on metabolites in cows”?Because only cows received had a good indicator.
Line 43-44: I am not sure what you want to say for the last sentence of the Abstract. The logic is not suitable for me. If you did not provide niacin, then the active form of niacin could not evaluate the biological effect, so how could it be a good indicator, as you stated previously. It is confusing here.
Line 55: could you please provide the general “biological proportion”.
Line 59: Please change “food” with “feed”
Line 70: could you please explain what is the physiological niacin?
Line 97: Please explicit what “NA” and “NAM” stand for
Line 102: Please change “grams” with “g”. Please be consistent
Line 103: what is the biological effect?
Line 127: how the cows were divided into two groups, based on which criteria?
Line 135: what was the placebo provided for control cows for the 120 g?
Line 144: why the blood samples were not taken before applying niacin? How and two weeks after you applied niacin before calving? How did you control the individual variation when you performed the statistical analysis?
Line 151: Please provide more details of the kit as presented in line 153.
Line 160: what is the ration?
Line 165: please explain the information that AUCROC could provide?
Line 173: Please remove the space after “NAD:”. Please be consistent with this, since before you didn’t use the space after “NAD:”
Line 175: Please remove the space after “NAD:”
Line 178: Please remove the space after “NAD:”
Line 179: Please change “statistically significant positive correlation” with “positive correlation”
Line 180: Please remove the space after “NAD:”
Line 180: Please change “ration” with “ratio”
Line 179-181: “while NAD: NADP ration increases with increasing NAD, and decreases with increasing NADP” this sentence is redundant since it is clear that the ratio increases when the numerator increases and so on. Please remove it.
Line 194: Please remove the space after “NAD:”
Line 196: Please change “food” with “feed”
Line 198: Please add “was” after “NAD”
Line 204: Please remove the space after “GLU:”
Line 204: Please change “shows” with “showed”
Line 207: Please change “cow shows” with “cows showed”. Please be consistent with the verb tense, since before you used the past
Line 210: Please add “was” after “NAD”
Line 248: Please change “food” with “feed”
Line 266: Please change “grams” with “g”
Line 272: what do you mean by “at the new”?
Line 278: Please change “food” with “feed”
Line 285: Please change “grams” with “g”
Line 287: Please insert a space between “12” and “g”
Line 288-289: Please change “were found to correlate with metabolites consistent” with “were correlated with metabolites consistently”
Line 292-293: Please remove “Therefore, it can be expected that there will be certain correlations.”
Line 300: Please change “on” with “in”
Line 306: Please change “reduce” with “reducing”
Line 312: Please change “grams” with “g”
Line 315: Please change “suggest” with “suggested”
Line 321: Please change “state” with “stated”
Line 323: Please change “state” with “stated”
Lines 341-342: with “nicotinmide” you mean “nicotinamide”?
Line 348: Please change “indicate” with “indicated”
Line 360-361: Please change “Almost all cows in the experimental group had a NAD: NADP ratio higher than 2 in only cows that received niacin.” with “Almost all cows in the experimental group that received niacin had a NAD: NADP ratio higher than 2.”
Line 364: Please remove the space after “NAD:”
Line 410: Please remove the space after “NAD:”
Line 410: Please remove “statistically”
Line 415: Please insert “.” after “applied niacin”
Table: It is not clear what is “Correlation (R2)”, how this analysis was performed and how all the ratios were calculated (etc. NAD: NADtoNADPratio”, what is that meaning?)
Fig 1: it is not clear to see the groups in the figure
Fig 2: the title is not matched to the figures. Please be consistent with the words used in both statistics and the title.
Table 2: please also provide the standard deviation or standard error for each correlation. Please not capitalize metabolic parameters and explain each metabolic parameter.
Fig 3: the information has been provided by Table 2, and the text is too small to read. The figures showed that it is crucial to measure the background NAD concentration before applying niacin.
Please check for the rest figures and tables after table 2.
Reviewer 3 Report
Dear author,
It is an interesting study and offers some interesting observations in the relationship between niacin status and metabolic parameters in dairy cows during early lactation.
However, there is too little data in the manuscript, for example, data on the DMI and lactation performance are missing. The introduction and discussion are lengthy and illogical. In addition, what’s the concentration of niacin in the basal diet? In my opinion, the current status of this manuscript is not suitable for publication in Animals.
Jian-bo Cheng